# APOBEC3B reporter myeloma cell lines identify DNA damage response pathways leading to APOBEC3B expression

Hiroyuki Yamazaki[1], Kotaro Shirakawa[1], Tadahiko Matsumoto[1], Yasuhiro Kazuma[1], Hiroyuki Matsui[1], Yoshihito Horisawa[1], Emani Stanford[1], Anamaria Daniela Sarca[1], Ryutaro Shirakawa[2], Keisuke Shindo[1], Akifumi Takaori-Kondo[1] *

1 Department of Hematology and Oncology, Graduate School of Medicine, Kyoto University, Kyoto, Japan,
2 Department of Molecular and Cellular Biology, Institute of Development, Aging and Cancer, Tohoku University, Sendai, Japan

* atakaori@kuhp.kyoto-u.ac.jp

**Data Availability Statement:** All relevant data are within the manuscript and its Supporting Information files.

## Abstract

Apolipoprotein B mRNA-editing enzyme catalytic polypeptide-like (APOBEC) DNA cytosine deaminase 3B (A3B) is a DNA editing enzyme which induces genomic DNA mutations in multiple myeloma and in various other cancers. APOBEC family proteins are highly homologous so it is especially difficult to investigate the biology of specifically A3B in cancer cells. To easily and comprehensively investigate A3B function in myeloma cells, we used CRISPR/Cas9 to generate A3B reporter cells that contain 3×FLAG tag and IRES-EGFP sequences integrated at the end of the A3B gene. These reporter cells stably express 3xFLAG tagged A3B and the reporter EGFP and this expression is enhanced by known stimuli, such as PMA. Conversely, shRNA knockdown of A3B decreased EGFP fluorescence and 3xFLAG tagged A3B protein levels. We screened a series of anticancer treatments using these cell lines and identified that most conventional therapies, such as antimetabolites or radiation, exacerbated endogenous A3B expression, but recent molecular targeted therapeutics, including bortezomib, lenalidomide and elotuzumab, did not. Furthermore, chemical inhibition of ATM, ATR and DNA-PK suppressed EGFP expression upon treatment with antimetabolites. These results suggest that DNA damage triggers A3B expression through ATM, ATR and DNA-PK signaling.

## Introduction

The apolipoprotein B mRNA-editing enzyme catalytic polypeptide-like DNA cytosine deaminase 3 family (APOBEC3, A3) consists of seven proteins (A3A, A3B, A3C, A3D, A3F, A3G and A3H) that preferentially induce C to U mutations in single strand DNA. A3 proteins were originally identified as factors of the innate immunity due to their mutagenic activity on viral genomes, and have recently joined the growing list of key intrinsic mutagens that play a part in oncogenesis [1]. Evidence for A3 mutagenicity consists of the presence of their mutational signature in cancer genomes [2], the effects observed when overexpressed in tumor tissues [3, 4],

**Funding:** This work was partly supported by JSPS KAKENHI Grant numbers JP19H03502, 18H03992 for A.T-K., JP19K07591 for K.S. and by Amedisys Home Health and Hospice Care (AMED) under Grant Number JP19ck0106250, JP19cm0106501 for A.T-K. Research funding from Ono Pharmaceutical Co. for A.T-K. The funders had no role in study design, data collection and analysis, decision to publish, or preparation of the manuscript.

**Competing interests:** The authors have declared that no competing interests exist.

as well as the correlation of APOBEC signature mutations with poor prognosis [5, 6]. Nevertheless, the precise biology of each APOBEC3 protein in cancer cells remains unknown. Due to the high structural homology of APOBEC3 family members, it is particularly difficult to obtain high-affinity- and high-specificity- antibodies against each APOBEC3 protein, which limits our capability to distinguish the precise role of each endogenous APOBEC3 during tumorigenesis.

Among APOBEC3s, we previously reported that endogenous A3B is overexpressed and seems to be the main source of deamination activity in most of the myeloma cell lines we examined [7]. Notably, high levels of A3B expression in tumor cells were an independent risk factor for the overall survival of myeloma patients [7] as well as of other cancer patients [8–11]. However, the regulatory mechanisms that mediate A3B expression have not been well studied. To date, molecules including cell cycle pathway [12] and DNA damage response (DDR) [13, 14] factors and several transcription factors such as human papillomavirus E6/E7 [15, 16], NF-κB [17, 18], c-Maf [5] and B-Myb [19] were reported to enhance A3B expression. Nevertheless, how these factors mediate A3B expression and how A3B contributes to tumor progression and/or acquisition of chemoresistance in myeloma cells remains unclear. To investigate A3B-associated myeloma biology, we used the CRISPR/Cas9 system to introduce the 3×FLAG tag and the IRES–EGFP gene at the beginning of the 3' UTR of the A3B gene in three human myeloma cell lines. We utilized this reporter cell lines to screen for how A3B expression is affected by anticancer treatments. Overall, we found these reporter cell lines to be very useful for the comprehensive analysis of A3B biology.

## Materials and methods

### Human cell lines and culture

Three human myeloma cell lines, U266, RPMI8226 and AMO1 cells were maintained in RPMI1640 (Nacalai) containing 10% FBS and 1% PSG (Invitrogen). HEK293T and Lenti-X cells were maintained in DMEM (Nacalai) containing 10% FBS and 1% PSG (Invitrogen).

### sgRNA design and construction of A3B reporter donor DNA

To design the single-guide RNA (sgRNA), the mRNA sequence of APOBEC3B (APOBEC3B Homo sapiens chromosome 22, GRCh38 Primary Assembly mRNA variant1, Fig 1A) was imported into CRISPRdirect [20]. After a target site was determined, annealed oligos (S1 Table) were inserted into pSpCas9(BB)–2A–Puro (PX459) V2.0 (Addgene, #62988) using the *BbsI* (New England Biolabs) cloning site, or into lentiCRISPR ver.2 (Addgene, #52961) using the *BsmBI* (New England Biolabs) cloning site as previously described [21, 22]. For the construction of the donor DNA vector (Fig 1B), the right homology arm, the modified cassette including the 3×FLAG–IRES–EGFP gene and the left homology arm were PCR-amplified using KOD FX Neo (ToYoBo). Each PCR primer pair contained around 15 bp overlaps. All the amplicons were cloned into the lentiviral plasmid pCSII–CMV–MCS (RIKEN, RDB04377) by using the In-Fusion HD Cloning Kit (TaKaRa), to produce the pCSII–CMV: A3B–3×FLAG–IRES–EGFP donor DNA plasmid (Fig 1C).

### Validation of sgRNA targeting efficiency

293T cells were transfected with pSpCas9(BB)–2A–Puro:sgRNA #4 (0.5 μg) using the FuGENE HD Transfection Reagent (Promega). Two days after transfection, 293T cells were harvested and their genomic DNA extracted using the QuickGene DNA whole blood kit S (KURABO). The targeted region was PCR-amplified from genomic DNA using the targeting test primers

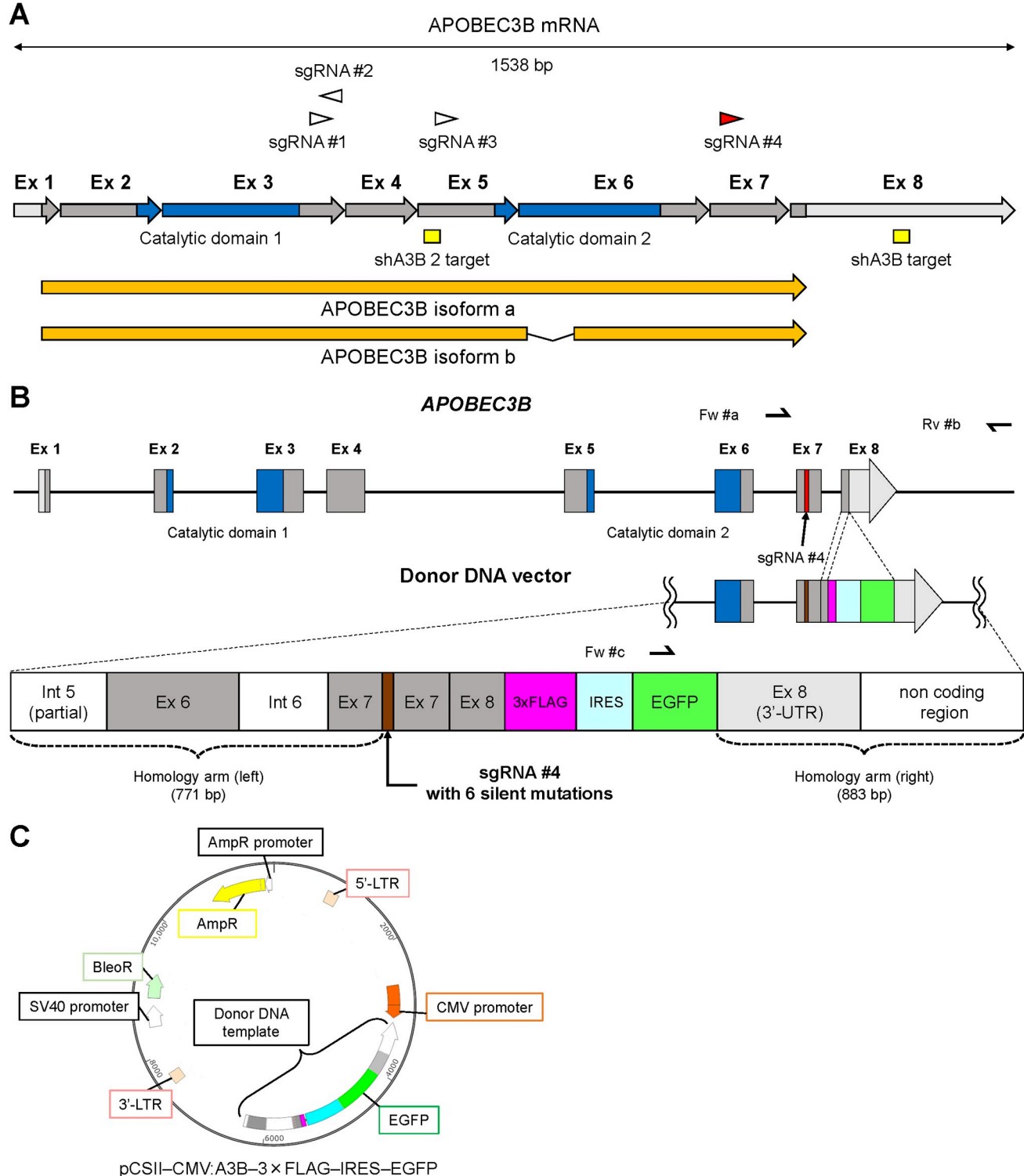

**Fig 1. Schema of APOBEC3B editing strategy.** (**A**) Schema of A3B mRNA structure. Triangles indicate highly-specific sgRNA target sites within *A3B*. Each arrow represents an exon (Ex). Areas in light gray show UTRs, those in dark gray show coding sequence regions (CDRs), and those in blue show catalytic domains. A3B mRNA isoforms (arrows in orange) as well as shA3B target sites (rectangles in yellow) are also indicated. (**B**) Schema of *A3B* in the host genome and in the donor DNA template. The donor DNA template contains six silent mutations in the sgRNA #4 target site, and intron 7 was removed. The 3×FLAG–IRES–EGFP sequence was inserted adjacent to the beginning of 3' UTR. (**C**) Schema of donor DNA plasmid, pCSII–CMV:A3B–3×FLAG–IRES–EGFP.

(S1 Table). The PCR products (200 ng) were denatured and then re-annealed to form hetero-duplex DNA. The hybridized DNA was digested with T7 endonuclease I (T7E1, New England Biolabs), and run on 2% agarose gel. Mutation frequency was calculated based on band intensity, using Image J software, as previously described [23].

### Generation of A3B reporter cell lines

For the U266 and AMO1 cell lines, $5 \times 10^6$ cells were co-transfected with 5 μg of pSpCas9 (BB)–2A–Puro:sgRNA #4 plasmid and 5 μg of pCSII–CMV:A3B–3×FLAG–IRES–EGFP donor DNA plasmid using the Amaxa Nucleofector (Lonza) with nucleofection solution R, program X-001. For the RPMI8226 cell line, $5 \times 10^6$ cells were transduced with lentiCRISPR ver.2:sgRNA #4 viruses and pCSII–CMV:A3B–3×FLAG–IRES–EGFP donor DNA viruses, simultaneously. These lentiviruses were produced by co-transfection of the packaging plasmid pVSVg (Addgene, #8454), psPAX2-D64V (Addgene, #63586) and lentiCRISPR ver.2:sgRNA #4 plasmid, or pCSII–CMV:A3B–3×FLAG–IRES–EGFP donor DNA plasmid, into Lenti-X cells.

### Flow cytometry analysis

Myeloma cells were stained with DRAQ7 (Biostatus) to mark dead cells, then were read on BD FACS Calibur or BD FACS Lyric (Becton-Dickinson Biosciences). To isolate A3B reporter cell lines, EGFP positive cells were sorted using a FACS Aria III cell sorter (Becton-Dickinson Biosciences) at seven days after transfection or transduction. The data was analyzed using the software FCSalyzer ver. 0.9.15-alpha. (https://sourceforge.net/projects/fcsalyzer/).

### Genotyping of A3B reporter single cell clones

Single cell clones were isolated from the sorted EGFP-positive cells of the three myeloma cell lines by limiting dilution. These clones were then PCR-genotyped using 2 pairs of the target confirmation primers, forward #a and reverse #b, and forward #c and reverse #b. To confirm the full sequence of A3B–3×FLAG–IRES–EGFP mRNA from the established cell line, complementary DNA (cDNA) was synthesized as described below, and was PCR-amplified by KOD FX Neo (ToYoBo) using a pair of primers, forward #d and reverse #e. The PCR products were sequenced using the 3130xl Genetic Analyzer (Applied Biosystems). All primers for PCR are listed in S1 Table.

### Immunoblot analysis

Whole cell lysates from $5.0 \times 10^6$ cells, prepared using an SDS-based buffer (5 mM EDTA, 1% SDS) supplemented with Protease inhibitor cocktail (Roche) and PhosSTOP EASY (Roche), were mixed with an equal volume of twofold concentrated sample buffer (Bio-Rad Laboratories) containing β-mercaptoethanol (Nacalai Tesque), and were treated for 5 min at 100˚C. Immunoblot analysis was performed as described previously using a mouse anti-FLAG antibody (Millipore, clone JBW301) or a mouse anti-α-tubulin monoclonal antibody (AA13, Funakoshi).

### Immunofluorescence assays

Cells were air-dried and fixed in 4% paraformaldehyde in phosphate-buffered saline (PBS) for 20 minutes on glass slides using Shandon cytospin 2 (THERMO FISHER SCIENTIFIC). Fixed cells were permeabilized, reduced and denatured for 30 minutes in PBS buffer containing 0.5% SDS, 5% β-mercaptoethanol and 10% FBS. Then, cells were washed three times with PBS

containing 4% FBS and 0.1% Triton X-100 (PFT buffer) [24], and incubated with a purified mouse anti-FLAG antibody for 1 hour. Subsequently, cells were incubated with a goat anti-mouse IgG (H+L)-Alexa Flour® 594 preadsorbed antibody (Abcam, ab150120) for 30 min in the dark. All antibodies were diluted with 3% BSA and 0.5% Tween in PBS. Then, the cells were stained with DAPI and were observed with a confocal laser scanning microscope (TCS-SP8, Leica).

## Knockdown experiments

We constructed pSicoR-mCherry lentiviral vectors [25] expressing short-hairpin RNA (shRNA) against A3B by inserting synthetic double-stranded oligonucleotides, as previously described [7] (TRCN0000140546 [26], sense oligo, 5′-TGCAAAGCAATGTGCTCCTGATCTCGAGATCAGGA GCACATTGCTTTGCTTTTTTC-3′, and antisense oligo, 5′-TCGAGAAAAAAGCAAAGCAATG TGCTCCTGATCTCGAGATCAGGAGCACATTGCTTTGCA-3′; TRCN0000139463, sense oligo, 5′-TCCTGATGGATCCAGACACATTCTCGAGAATGTGTCTGGATCCATCAGGTTTTTTC-3′, and antisense oligo, 5′-TCGAGAAAAAACCTGATGGATCCAGACACATTCTCGAGAATGTGTC TGGATCCATCAGGA-3′) into the cloning site. For non-target shRNA, we used two constructs that were cloned as scrambled sequences (control [27], sense oligo, 5′-TGTCAAGTCTCACTT GCGTCTTCAAGAGAGACGCAAGTGAGACTTGACTTTTTTC-3′, antisense oligo, 5′-TCGAGA AAAAAGTCAAGTCTCACTTGCGTCTCTCTTGAAGACGCAAGTGAGACTTGACA-3′; control-2 [28], sense oligo, 5′-TATCTCGCTTGGGCGAGAGTAAGCTCGAGCTTACTCTCGCCCAAGCG AGATTTTTTTC-3′, antisense oligo, 5′-TCGAGAAAAAAATCTCGCTTGGGCGAGAGTAAGC TCGAGCTTACTCTCGCCCAAGCGAGATA). The lentivirus was produced by co-transfection of Trans-Lentiviral packaging plasmid mix (GE Dharmacon) and pSicoR-mCherry into Lenti-X cells.

## Quantitative RT-PCR

Total RNA was extracted from cell lines using the High Pure RNA isolation kit (Roche). cDNA was synthesized using the PrimeScriptR II 1st strand cDNA Synthesis Kit (Takara) by random primer and oligo dT primer mixture. Real-time PCR was performed using the Thunderbird SYBR qPCR Mix (ToYoBo). Target gene expression levels were normalized by endogenous expression levels of HPRT1. All primers for real-time PCR are listed in S1 Table.

## Anticancer treatment screening

To examine the effects of chemotherapeutic agents on A3B expression, the A3B reporter cells were cultured for two days at a concentration of $2 \times 10^5$ cells/well/1.5 mL medium in 12-well plates and treated with phorbol 12-myristate 13-acetate (PMA, Sigma), melphalan (MEL, Wako), cisplatin (CDDP, Nihon-kayaku), mitomycin C (MMC, Funakoshi), N-desacetyl-N-methylocolchicine (COL, KaryoMAX Colcemid Solution in PBS, Thermo Fisher), camptothecin (CPT-11, TopoGEN), etoposide (VP-16, TREVIGEN), cytosine-1-B-D(+)-arabinofuranoside (Ara-C, Wako), gemcitabine hydrochloride (GEM, Sigma), hydroxyurea (HU, Tokyo chemical industry), aphidicolin (APH, Wako), bortezomib (BOR, Funakoshi), lenalidomide (LEN, Sigma), elotuzumab (ELO, Bristol-Myers Squibb), human IFN-α (Sumiferon, Dainippon Sumitomo Pharma) or olaparib (Funakoshi) at several concentrations as described in the main text. These chemotherapeutics were dissolved in 100% dimethyl sulfoxide (Nacalai Tesque) with the exception of COL, HU, ELO and INF-α which were dissolved in distilled water. To examine the effects of radiation or UV on A3B expression, the cells were exposed to gamma radiation using a Cs-137 Gamma Cell or to UVC using a FUNA UV Crosslinker, FS-800 (Funakoshi). To examine the effects of kinase inhibitors on A3B regulation, KU-55933

(Selleck), VE-821 (Selleck), NU-7026 (Selleck) or CGK733 (Calbiochem) were added 2 hours prior to antimetabolite treatment.

## Statistical analysis

Mann-Whitney U test and Welch's t test were calculated to evaluate the differences in continuous variables between two groups for quantitative RT-PCR results and flow cytometry results, respectively, by using the EZR software (version 3.0.2, Saitama Medical Center, Jichi Medical University) [29].

# Results

## CRISPR design

We first designed sgRNA candidates for target sites in *A3B*, excluding introns, using the web based tool, CRISPRdirect [20]. There are only four highly specific candidates for *A3B* (Fig 1A and S2 Table) mainly due to the high homology among APOBEC3 family genes. In order to insert the 3×FLAG sequence into *A3B* with a minimal off-target effect, we selected sgRNA #4 (Fig 1A). U266, RPMI8266 and AMO1 endogenously overexpress *A3B* [7]. We used the pSpCas9(BB)–2A–Puro plasmid and the lentiCRISPR ver.2 plasmid to transduce the CRISPR system that targets *APOBEC3B* loci in these cell lines. We also used a donor DNA template to introduce the 3×FLAG and IRES-EGFP reporter sequences at the end of the coding region and to move the stop codon behind the 3×FLAG sequence (Fig 1B and 1C). The 3×FLAG–IRES–EGFP cassette was located adjacent to the beginning of 3' UTR, and intron 7 (281bp) was removed to prevent it from becoming the right homology arm. Usually, the PAM sequence (NGG) in the donor DNA template must be mutated to prevent cutting by Cas9, however, in our case, any mutation of PAM would lead to alteration of the A3B protein sequence. Instead, we designed six silent mutations within the target site to inhibit efficient future sgRNA binding: the host genomic target sequence of sgRNA #4, 'ctgGGACACCTTTG TGTACCGCCAGGgat', was altered to 'ctgGGACAC̲GTT̲C̲GTC̲TAT̲CGAC̲AA̲Ggat', in the donor DNA template sequence. Finally, the complete donor DNA template sequence was inserted into the pCSII–CMV–MCS plasmid in the opposite direction of the CMV promoter of the parental vector (Fig 1C), so that cells in which the donor DNA vector is present merely transiently would not express *EGFP* and only cells which had their genome successfully engineered would emit EGFP fluorescent signals.

## CRISPR guided 3×FLAG–IRES–EGFP insertion in *A3B* locus

To test the targeting efficiency of the sgRNA, we transfected the pSpCas9(BB)–2A–Puro: sgRNA #4 plasmid into 293T cells. The transfection efficiency was 15.2% (T7E1 assay, Fig 2A), therefore, we proceeded to co-transfect/co-transduce Cas9, the sgRNA #4 expressing vector and the donor DNA vector into U266, RPMI8226 and AMO1 cell lines. As expected, the efficiency of genome editing in myeloma cells was quite low, but we successfully enriched EGFP positive cells by cell sorting (Fig 2B). Single clones were isolated by limiting dilution from each cell line, expanded and A3B genotype was confirmed by PCR. Out of all the isolated clones, we selected the following four edited cell lines: U266[A3B–3×FLAG–IRES–EGFP] #1 and #2 (U266 KI #1 and #2), RPMI8226[A3B–3×FLAG–IRES–EGFP] (RPMI8226 KI), and AMO1[A3B–3×FLAG–IRES–EGFP] (AMO1 KI). According to the genotype PCR in Fig 2C, the A3B–3×FLAG–IRES–EGFP cassette was correctly integrated at the target site in these cell lines. Of note, both A3B alleles in U266 KI #2 were edited (Fig 2C). To confirm the mRNA sequence of A3B–3×FLAG–IRES– EGFP, we PCR-amplified the full length of the cDNA derived from each cell line (Fig 2D) and

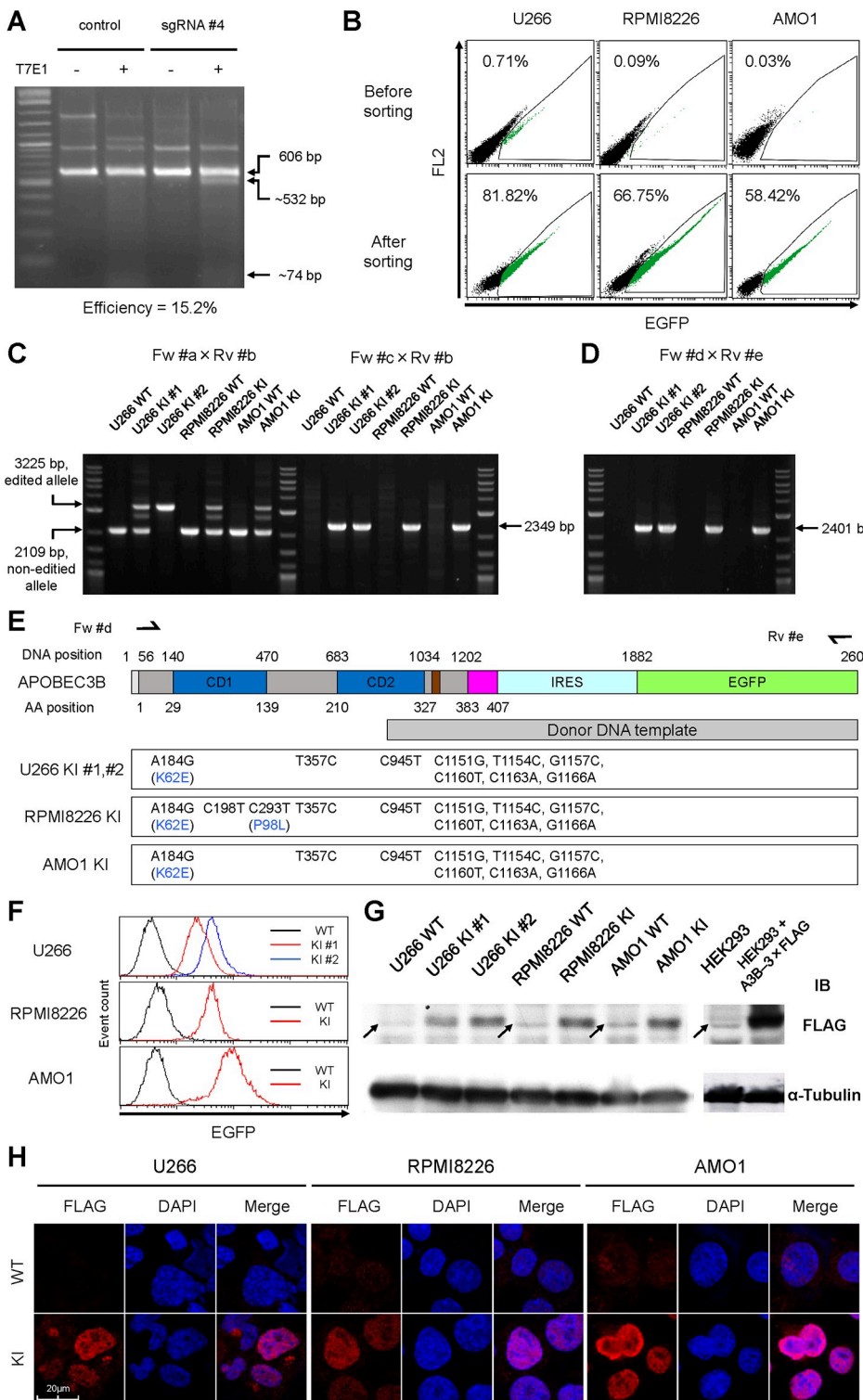

**Fig 2. Establishment of A3B–3×FLAG–IRES–EGFP knock-in myeloma cell lines.** (**A**) T7E1 assay of the sgRNA #4 site in 293T cells. Expected positions of uncleaved (606 bp) and cleaved (532 bp and 74 bp) DNA bands by T7E1 are indicated with arrows. The mutation frequency is also shown. (**B**) Flow cytometry of U266, RPMI8226 and AMO1 cells after introducing the donor DNA vector along with the CRISPR-Cas9 vector. EGFP positive cells are highlighted in green and their proportions are indicated. (**C**) Genotyping PCR of genomic DNA from each clone derived from a single cell among the enriched cells in (B). Each clone was genotyped by two pairs of primers, Fw #a × Rv #b and Fw

#c × Rv #b. Using the former primer pair, the expected size of the PCR amplicon is 2109 bp for the wild type allele, and 3225 bp for the knock-in allele. Using the latter primer pair, the PCR amplicon (2349 bp) can be detected only if the knock-in allele is present. (**D**) Genotyping PCR of cDNA from each clone in (C). The PCR amplicon (2401 bp) can be detected only if the knock-in allele is present. (**E**) Sanger sequencing results for the full length of the edited A3B cDNA originated from the clones of 3×FLAG–IRES–EGFP knock-in cell lines. Schema of the A3B–3×FLAG–IRES–EGFP mRNA structure is also depicted, the same as in Fig 1B. (**F**) Histograms of EGFP intensity values from the 3×FLAG–IRES–EGFP knock-in cell lines as determined by flow cytometry. (**G**) Immunoblot analysis of the 3×FLAG–IRES–EGFP knock-in cell lines. Lysates of untransduced HEK293 and A3B–3×FLAG overexpressing HEK293 were also blotted as negative and positive control, respectively. Arrows indicate non-specific bands. α-tubulin was evaluated as internal control. (**H**) Immunofluorescence analysis of the 3×FLAG–IRES–EGFP knock-in clones using an anti-FLAG antibody. For U226 KI, clone U266 KI #1 was examined. Images were obtained by confocal fluorescence microscopy (magnification, x630).

performed Sanger sequencing analysis. As desired, all the engineered cell lines possessed correct A3B–3×FLAG sequences, including the intended 6 silent mutations in the sgRNA target site and SNPs in the unmanipulated region (Fig 2E). According to flow cytometry analysis, the intensity of the fluorescent signal increased in the order of U266 KI #1, RPMI8226 KI and AMO1 KI, which is consistent with their A3B expression levels in a previous report [7]. U266 KI #2 exhibited around two times stronger fluorescence than U266 KI #1, indicating that the 3×FLAG–IRES–EGFP gene was integrated homozygously in U266 KI #2 and heterozygously in U266 KI #1. According to the results of flow cytometry and PCR-genotyping (Fig 2C), RPMI8226 KI and AMO1 KI contain a single allele of the 3×FLAG–IRES–EGFP gene. Immunoblot analysis also confirmed that all the cell lines produced A3B–3×FLAG proteins of the predicted size (Fig 2G). Immunofluorescent analysis of the subcellular localization of A3B–3×FLAG proteins showed a dominant localization in the nucleoplasm (Fig 2H), which is identical with that of wild type A3B proteins [7].

## The established A3B–3×FLAG–IRES–EGFP knock-in cell lines work as A3B reporters

To verify the feasibility of the established cell lines as A3B reporters, we first transduced RPMI8226 KI and AMO KI cells with lentiviral shRNA against A3B together with an EF1α-driven mCherry fluorescent marker. When A3B mRNA was efficiently depleted (Fig 3A), A3B–3×FLAG protein levels decreased as expected (Fig 3B). Similarly, EGFP fluorescence intensity decreased in mCherry positive, shRNA transduced cells, compared with mCherry negative, shRNA non-transduced cells (Fig 3C–3F). Next, we treated U266 KI #1, RPMI8226 KI and AMO1 KI cells with PMA, a PKC activator, which is known to upregulate A3B expression via the NF-κB pathway [17, 18]. A quantitative RT-PCR analysis confirmed the enhancement of A3B mRNA levels for each cell line (Fig 3G). Consistently, immunoblot analysis detects increases of A3B–3×FLAG proteins (Fig 3H), and flow cytometry analysis detects peak shifts and increases of mean fluorescent intensity (MFI) for each cell line (Fig 3I and 3J). Based on the above results, we conclude that these established cell lines can be used as reliable A3B reporters.

## DDR upregulates A3B expression via all the DDR-PIKK pathways in myeloma cells

Because the established A3B reporter cell lines provide an easy way to evaluate the alteration of A3B expression by simply performing flow cytometry analysis, we investigated which of the current clinically approved myeloma treatments affect A3B expression. Interestingly, most conventional anticancer treatments which induce DNA interstrand cross-links (e.g., CDDP, MEL and MMC), microtubule inhibition (e.g.,COL), topoisomerase inhibition (e.g.,CPT-11

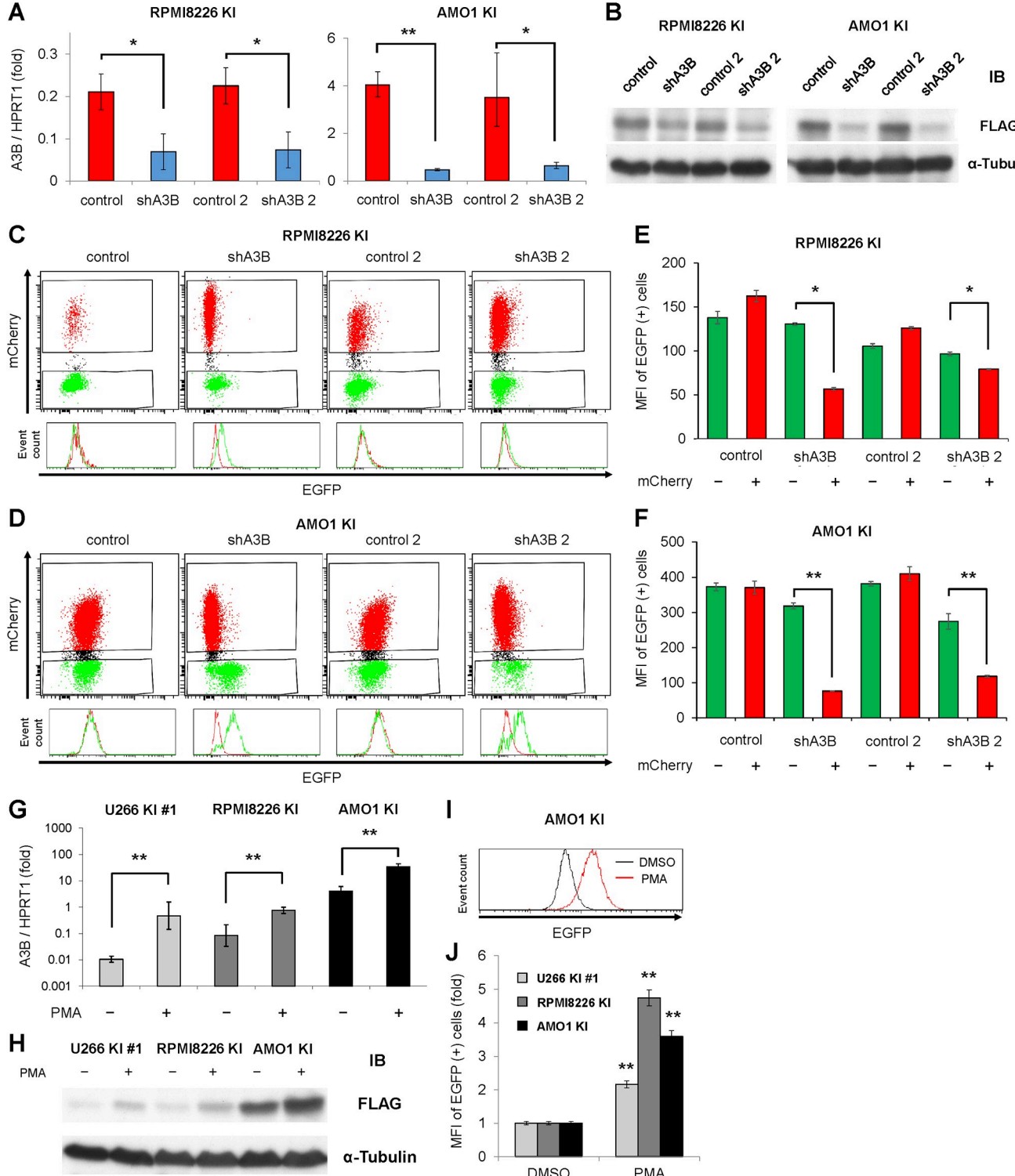

**Fig 3. A3B–3×FLAG–IRES–EGFP knock-in cells work as A3B reporters.** (**A**, **B**) Real-time PCR (A) and immunoblotting (B) of A3B in RPMI8226 KI cells and AMO1 KI cells, which were transduced with lentiviral shRNA against A3B (two constructs: shA3B or shA3B-2) or control (two constructs: control or control-2). HPRT1 or α-tubulin were evaluated as internal controls. Mann-Whitney U tests were used to compare the results between control and A3B knockdown samples: $^{**}P < 0.01$; $^{*}P < 0.05$. (**C**, **D**) Flow cytometry of RPMI8226 KI cells (C) and AMO1 KI cells (D) at 17 days after transduction with lentiviral shRNA against A3B or control. In the histogram representation, EGFP intensity was compared between mCherry positive cells (colored in red) and mCherry negative cells (colored in green). (**E**, **F**) Bar graph of EGFP mean fluorescence intensity (MFI) of the experiments in Figures (C, D).

Mann-Whitney U tests were performed to compare the results between mCherry negative and mCherry positive samples: **P < 0.01; *P < 0.05. (**G, H**) Real-time PCR (G) and immunoblotting (H) of A3B in three A3B–3×FLAG–IRES–EGFP knock-in cell lines, which were treated with PMA (20 ng/mL) for 6 hours and 24 hours, respectively. Mann-Whitney U tests were performed to compare the results between control and PMA-treated samples: **P < 0.01. (**I**) Representative result of EGFP intensity histogram of AMO1 KI cells, which were treated with PMA (20 ng/mL) for 2 days. (**J**) Bar graph of EGFP mean fluorescent intensity (MFI) of three A3B–3×FLAG–IRES–EGFP knock-in cell lines, which were treated with PMA (20 ng/mL) for 2 days. Mann-Whitney U tests were performed to compare the results between control and PMA-treated samples: **P < 0.01.

and VP-16), DNA synthesis inhibition (e.g., Ara-C, GEM, HU and aphidicolin) or DNA double-strand breaks (e.g., radiation), exacerbated endogenous A3B overexpression (Fig 4A and 4B). Treatment with olaparib alone, a Poly(ADP-ribose) polymerase (PARP) inhibitor, which is known to induce SSBs that are degraded to DSBs during replication [30], also enhanced A3B expression (Fig 4C). On the other hand, the proteasome inhibitor (i.e., BOR), the immunomodulatory drug (i.e., LEN), the non-agonistic antibody drug (i.e., ELO) and INF-α did not enhance A3B expression levels (Fig 4A). These results intimate that DNA toxic stimulation upregulates A3B expression through DDR and following activation of DDR associated phosphatidylinositol 3' kinase-related kinases (DDR-PIKKs) [31] including ataxia telangiectasia and Rad3-related (ATR), and ataxia telangiectasia-mutated (ATM), DNA-dependent protein kinase (DNA-PK). Chemical inhibition of DDR-PIKKs by VE-821 for ATR, or NU-7026 for DNA-PK, suppressed EGFP increase upon antimetabolite treatment (Fig 4D and 4E). Moreover, various combinations of PIKK inhibitors, including KU-55933, an ATM inhibitor, exhibited a synergistic effect of preventing A3B expression increase upon antimetabolite stimulation (Fig 4D and 4E). Notably, pretreatment with CGK733 alone, which was first reported as an ATM/ATR inhibitor [32], almost completely blocked the antimetabolite effect on A3B expression in the three cell lines (Fig 4F). These results suggest that all the DDR-PIKK pathways might be involved in A3B regulation in myeloma cells.

## Discussion

In the present report, we successfully established four A3B reporter cell lines derived from three human myeloma cell lines, U266, RPMI8226 and AMO1. These cell lines express EGFP proteins with attribution to A3B expression, regulated by the same transcriptional/posttranscriptional mechanisms due to identical promoter, 3'-UTR and 5'-UTR to *A3B*. Due to these particularities, these cell lines are a very useful tool for investigating A3B regulation in a high-throughput screening format by flow cytometry analysis, which will allow for the development of specific A3B suppressors. There are several similar reports of other gene-edited reporter cell lines used for comprehensively studying the transcriptional regulation of the targeted gene [33–37]. In the case of A3B, most previous reports have studied A3B protein function using exogenous overexpression by transient transfection in a limited number of cell lines including non-human cells [16, 26, 38–44], mainly due to the difficulty of obtaining specific anti-A3B antibodies. In contrast, the commercially available and certified anti-FLAG antibody can be used to explore the A3B protein in the established A3B reporter cell lines described here. That is to say, these cell lines have the potential to clarify natural protein-protein and/or DNA-protein interaction of A3B specifically, in tumor cells. In addition, the A3B reporter system can be integrated into other A3B-overexpressing cell lines by using the Cas9/sgRNA #4 expressing vector and pCSII–CMV:A3B–3×FLAG–IRES–EGFP donor DNA vector described here.

According to our pilot screening, most of the conventional anticancer treatments exacerbated A3B overexpression in myeloma cells (Fig 4A and 4B). These treatments seem to act through a common pathway: induction of DDR [45]. Specifically, HU, which inhibits the incorporation of nucleotides by interfering with the enzyme ribonucleotide reductase [46], and APH, which interferes with DNA replication by inhibiting DNA polymerases α, ε and δ

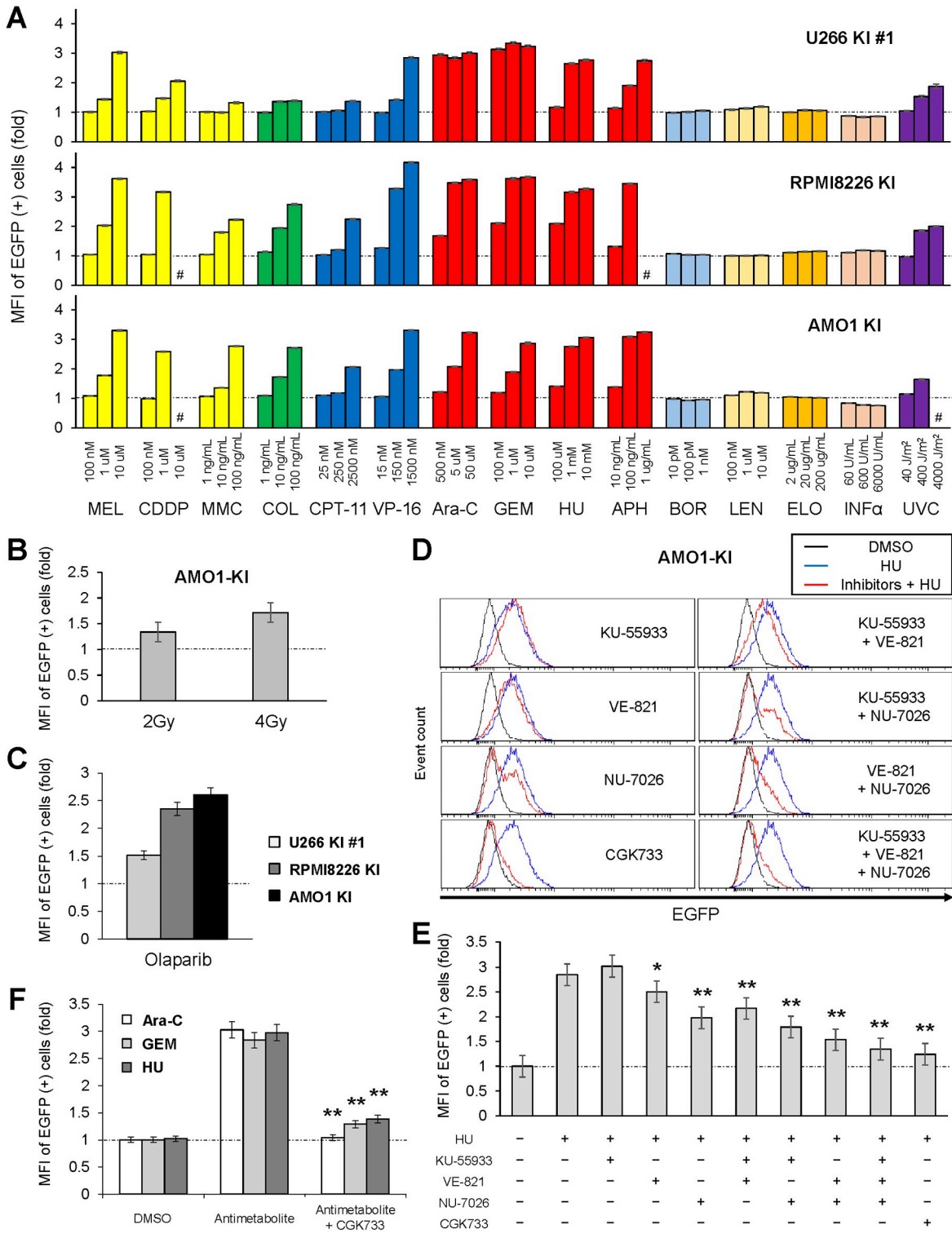

**Fig 4. DNA damage response exacerbates A3B overexpression via the DDR-PIKK pathways in myeloma cells.** (**A**) A panel of EGFP mean fluorescent intensity (MFI) of three A3B–3×FLAG–IRES–EGFP knock-in cell lines with various anti-cancer treatment. A3B reporter cells were incubated for 2 days with each anti-cancer reagent at the concentrations indicated on the horizontal axis. For the UVC exposure experiment, A3B reporter cells were irradiated with a single dose at 2 days before flow cytometry analysis. Hash mark (#) represents unmeasurable state due to cytotoxicity. (**B**) Bar graph of EGFP MFI of AMO1 KI cells, which were exposed to a single dose of γ-ray at 2 days before flow cytometry analysis. (**C**) Bar graph of EGFP MFI of three A3B–3×FLAG–IRES–EGFP knock-in cell lines with

olaparib treatment (10 μM) for 2 days. (**D, E**) Histograms (D) and bar graphs (E) of EGFP intensity values from AMO1 KI cells, which were co-treated with HU (1μM) and DDR-PIKK inhibitors: KU-55933, 5 μM; VE-821, 5 μM; NU-7026, 2 μM; CGK733, 5 μM. Cells were incubated with the reagents for 2 days and subsequently analyzed by flow cytometry. Mann-Whitney U tests were performed to compare the results between HU-treated and PIKK inhibitor-treated samples: $^{**}$P < 0.01; $^{*}$P < 0.05. (**F**) Bar graph of EGFP MFI of three A3B–3×FLAG–IRES–EGFP knock-in cell lines treated with an antimetabolite (Ara-C, 50 μM; GEM, 1 μM; HU, 1 μM) with or without CGK733 (5 μM) for 2 days. Mann-Whitney U tests were performed to compare the results between antimetabolite-treated and CGK733-treated sample: $^{**}$P < 0.01.

[47], are both commonly used to induce replication fork stalling that leads to ATR/ATM activation. These antimetabolites are also known to induce DSBs [48, 49]. CPT-11 covalently stabilizes the topoisomerase I–DNA cleavage complex by inhibiting the ligation of SSBs [50], thereby increasing the number of SSBs and subsequent DSBs [51]. Meanwhile, VP-16 leads to increases in the levels of topoisomerase II–DNA covalent complexes resulting in the rapid induction of DSBs [52]. DNA interstrand cross-linkers form a number of adducts with DNA, and thereafter activate a wide variety of DNA repair pathways [53] such as nucleotide excision repair (NER) [54, 55], homology-directed repair (HDR) [56] and mismatch repair (MMR) [57]. DNA interstrand cross-links are also known to be sensed by non-histone chromosomal high-mobility group box proteins 1 and 2 (HMGB1 and HMGB2), which affect cell cycle events and subsequently induce apoptosis [58]. Colcemid also has the potential to induce DSBs [59, 60]. Although UV exposure dominantly produces cyclobutane pyrimidine dimers (CPDs) and 6–4 photoproducts (6-4PP) but not DSBs directly, it activates ATR by SSBs and ATM by DSBs in a NER-dependent manner [61]. On the other hand, bortezomib and lenalidomide did not enhance A3B overexpression (Fig 4A). We cannot exclude the possibility that these drugs can directly cause DSBs, however, there are few reports of DNA damage induced by a single treatment with bortezomib or lenalidomide.

The upregulation of A3B expression induced by DNA damage was suppressed by DDR-PIKK inhibitors, consistent with a previous report in breast cancer [13]. Under single-inhibition of each DDR-PIKK pathway, the DNA-PK inhibitor (NU-7026) suppressed A3B elevation the strongest. Kanu et al. reported that inhibiting ATR, and to a lesser extent ATM, reduced hydroxyurea-induced A3B activation, and concluded that DNA replication stress activates transcription of A3B via an ATR/Chk1-dependent pathway in breast cancer [13]. Thus, the dependency of A3B regulation on each DDR-PIKK pathway could vary among cancer cell types. On closer examination of the histograms in our study, EGFP signal curves from cells treated with NU-7026 had two peaks in contrast to those treated with VE-821 which had only one peak (Fig 4D), suggesting that DNA-PK inhibition completely blocked A3B upregulation in a certain population of cells, whereas ATR inhibition suppressed it in all cells. Considering the synergistic effects of the combinations of DDR-PIKK inhibitors in our study (Fig 4D and 4E), it seems that all the DDR-PIKK pathways are at least partly involved in A3B regulation in myeloma cells. This model is also supported by the redundancy between DDR-PIKK pathways under DNA replication stress [62]. Interestingly, A3B induction by DDR was almost completely blocked by treatment with CGK733 alone. CGK733 was initially reported to inhibit both ATM and ATR kinase activities, however, its specificity is now considered controversial [63, 64]. Nonetheless, there seems to be no doubt that CGK733 targets at least partly a downstream factor of the ATM/ATR pathway [65, 66]. HMGB1 and Cdc7 were identified as new target kinase candidates of CGK733 [67]. Of note, proteasome inhibitors were reported to suppress DDR by inhibiting phosphorylation of DDR-PIKKs [68, 69]. This suppression effect could explain why bortezomib did not exacerbate A3B expression.

We previously reported that shRNA against A3B decreased the basal level of γH2AX foci in myeloma cell lines, indicating that A3B induces constitutive DNA double-strand breaks,

promoting DDR activation [7]. Therefore DDR-inducible treatments trigger a positive feedback loop for A3B expression, which may drive chemoresistant clone expansion during chemotherapy. To prevent disease progression and potentiate current therapy, conventional anticancer treatment coupled with a combination of DDR-PIKK inhibitors including a proteasome inhibitor might not only have a synergistic cytotoxicity for tumor cells but also suppress the production of chemoresistant clones.

## Supporting information

**S1 Fig. Original membranes and gels.**
(PDF)

**S1 Table. List of oligos and thermal cycle conditions for genotyping PCR.**
(XLSX)

**S2 Table. sgRNA target candidates for A3B.**
(XLSX)

## Acknowledgments

Plasmids including pSpCas9(BB)–2A–Puro (PX459) V2.0 (#62988), lentiCRISPR ver.2 (#52961), pVSVg (#8454), psPAX2-D64V (#63586) were provided by Addgene.

## Author Contributions

**Conceptualization:** Hiroyuki Yamazaki, Kotaro Shirakawa, Akifumi Takaori-Kondo.

**Funding acquisition:** Kotaro Shirakawa, Akifumi Takaori-Kondo.

**Investigation:** Hiroyuki Yamazaki, Kotaro Shirakawa, Tadahiko Matsumoto, Yasuhiro Kazuma, Hiroyuki Matsui, Yoshihito Horisawa, Emani Stanford, Anamaria Daniela Sarca, Keisuke Shindo.

**Methodology:** Hiroyuki Yamazaki, Ryutaro Shirakawa.

**Resources:** Hiroyuki Yamazaki, Ryutaro Shirakawa.

**Supervision:** Kotaro Shirakawa, Akifumi Takaori-Kondo.

**Validation:** Akifumi Takaori-Kondo.

**Writing – original draft:** Hiroyuki Yamazaki, Kotaro Shirakawa.

**Writing – review & editing:** Hiroyuki Yamazaki, Kotaro Shirakawa, Tadahiko Matsumoto, Yasuhiro Kazuma, Hiroyuki Matsui, Yoshihito Horisawa, Emani Stanford, Anamaria Daniela Sarca, Keisuke Shindo, Akifumi Takaori-Kondo.

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
