## [Decision Letter · Decision Letter 0]

9 Oct 2019

PONE-D-19-26273

APOBEC3B reporter myeloma cell lines identify DNA damage response pathways leading to APOBEC3B expression

PLOS ONE

Dear Dr. Takaori-Kondo,

Thank you for submitting your manuscript to PLOS ONE. After careful consideration, we feel that it has merit but does not fully meet PLOS ONE’s publication criteria as it currently stands. Therefore, we invite you to submit a revised version of the manuscript that addresses the points raised during the review process.

Both reviewers raised minor but important points that should be addressed. These include both editorial and content points, requiring both textual and experimental revisions. Further, there was a small concern regarding grammar and statistical analysis. In addition, it was mentioned that only a single clone was analyzed. Additional clones would add significant value. 

We would appreciate receiving your revised manuscript by Nov 23 2019 11:59PM. To enhance the reproducibility of your results, we recommend that if applicable you deposit your laboratory protocols in protocols.io, where a protocol can be assigned its own identifier (DOI) such that it can be cited independently in the future. For instructions see: http://journals.plos.org/plosone/s/submission-guidelines#loc-laboratory-protocols

We look forward to receiving your revised manuscript.

Kind regards,

Robert W Sobol, PhD

Academic Editor

PLOS ONE

Journal Requirements:

Additional Editor Comments (if provided):

Reviewers' comments:

Reviewer's Responses to Questions

**Comments to the Author**

1. Is the manuscript technically sound, and do the data support the conclusions?

Reviewer #1: Yes

Reviewer #2: Yes

2. Has the statistical analysis been performed appropriately and rigorously? 

Reviewer #1: No

Reviewer #2: I Don't Know

3. Have the authors made all data underlying the findings in their manuscript fully available?

Reviewer #1: Yes

Reviewer #2: Yes

4. Is the manuscript presented in an intelligible fashion and written in standard English?

Reviewer #1: Yes

Reviewer #2: Yes

5. Review Comments to the Author

Reviewer #1: APOBEC3B reporter myeloma cell lines identify DNA damage response pathways leading to APOBEC3B expression.

Authors use CRSPR technology to investigate the APOBEC3B protein. The methodology and diagram made the genetic engineering steps easy to follow for the reader. The authors show a good understanding of the CRISPR insertion protocol.

Minor points:

1. Line 215- Can the author clarify the sentence, “and intron 7 was removed to avoid partial gene

editing.”

2. Line 234- The authors ability to only produce one clone in the RPMI8226 and AMO1 lines confounds the data for the reader. It is unclear whether the differences are related to changes in the single cell clone or the gene editing. This is compounded further with both homozygous and heterozygous integrations in the cell lines.

3. Line 250- In the reviewers experience the FLAG protein is very easy to detect, yet in the presented work the bands seem rather faint and there is also FLAG signal in the WT cell lines. Can the author comment on this?

4. Figure 2H – U266 FLAG distribution is not particularly convincing. Can the author provide a better example?

5. Figure 3C and 3D, Line 259- EGFP fluorescent intensity

260- decreased in mCherry positive, shRNA transduced cells, compared with mCherry negative,

261- shRNA untransduced cells (Fig. 3C and 3D). Can the author present this data in a different way, it is not clear to the reviewer how large the difference really may be?

6. Line 291- “These results suggest that all the DDR related pathways are involved in A3B regulation in myeloma cells.” This has not been testing in the presented study, there are dozens of DDR pathways, using a few ATR/ATM inhibitors does not prove the above statement.

Various spelling mistakes through-out manuscript.

Major points:

1. There is a lack of meaningful statistical analysis on many of the graphs making it difficult to determine whether the results are significant, single experiments or biologically relevant.

Reviewer #2: The authors demonstrate the creation of CRISPR/Cas9 knock in of 3x FLAG tag and IRES-EGFP to the end of APOBEC3B gene in myeloma cell lines that they subsequently use as a reporter for chemotherapeutic treatments that induce APOBEC3B expression. Using this system they were able to identify inhibition of ATM, ATR and DNA-PK suppressed EGFP expression in response to antimetabolite treatments in several myeloma cell lines. This work is important to the understanding of APOBEC3B in myeloma and the findings support what is already known in breast cancer regarding APOBEC3B and ATM/ATR inhibition.

The methods are well described and the reporter system well characterized in the figures.

Major issues: none

Minor issues: Methods section: Statistical analysis described but not reported anywhere in the text, figures, or figure legends. Suggest adding this information prior to publication.

6. PLOS authors have the option to publish the peer review history of their article (what does this mean?). If published, this will include your full peer review and any attached files.

Reviewer #1: No

Reviewer #2: No

---

## [Author Response · Author response to Decision Letter 0]

28 Nov 2019

Response to the Reviewers:

We thank the reviewers for their useful comments and suggestions. We have addressed all the issues they outlined and revised the manuscript as detailed below. Changes in the manuscript are highlighted in red. We asked native English speakers in the lab to proofread the manuscript again, corrected typos and grammatical errors and made changes so that readers can easily follow the manuscript. We also added acknowledgements for plasmids and research funding that we forgot to mention in the original manuscript. We hope the reviewers find that our manuscript has much improved, is more robust, easier to read and suitable for publication in PLOS ONE.

5. Review Comments to the Author

Reviewer #1: APOBEC3B reporter myeloma cell lines identify DNA damage response pathways leading to APOBEC3B expression.

Authors use CRSPR technology to investigate the APOBEC3B protein. The methodology and diagram made the genetic engineering steps easy to follow for the reader. The authors show a good understanding of the CRISPR insertion protocol.

Minor points:

1. Line 215- Can the author clarify the sentence, “and intron 7 was removed to avoid partial gene editing.”

We changed the sentence in Line 215 to “and intron 7 (281bp) was removed to prevent it from becoming the right homology arm”, because if the DNA repair after CRISPR cleavage happened using this region including intron 7 as right homology arm, only 6 silent mutations would be introduced and the reporter sequence could not be introduced.

2. Line 234- The authors ability to only produce one clone in the RPMI8226 and AMO1 lines confounds the data for the reader. It is unclear whether the differences are related to changes in the single cell clone or the gene editing. This is compounded further with both homozygous and heterozygous integrations in the cell lines.

We obtained multiple heterozygous clones for each cell line, and were fortunate to also find a clone in the U266 cell line that was homozygous for the edited A3B gene, so we decided to include it in our analysis. The two clones, U266 KI #1 and #2, exemplified how the GFP expression increases if both alleles are edited. To clarify this, we changed the lines 235-236 and 247.

3. Line 250- In the reviewers experience the FLAG protein is very easy to detect, yet in the presented work the bands seem rather faint and there is also FLAG signal in the WT cell lines. Can the author comment on this?

We added the immunoblots for A3B overexpressing 293T lysate as positive control and mock transfected 293T cell lysates as negative control in Fig 2G. We believe that the amount of the A3B protein overexpressed in myeloma cell lines was substantially small compared to that of A3B overexpressed in 293T by transfection of the A3B expressing plasmids. The extra band that the reviewer pointed out is a faint non-specific band that appears just below the A3B in myeloma cell lines and even in 293T cells. We put arrows for these non-specific bands in Fig. 2G. 

4. Figure 2H – U266 FLAG distribution is not particularly convincing. Can the author provide a better example?

We replaced the immunofluorescent images for Fig 2H, U266 for a better example.

5. Figure 3C and 3D, Line 259- EGFP fluorescent intensity

260- decreased in mCherry positive, shRNA transduced cells, compared with mCherry negative,

261- shRNA untransduced cells (Fig. 3C and 3D). Can the author present this data in a different way, it is not clear to the reviewer how large the difference really may be?

We added bar graphs of MFI values with error bars in Figure 3E and 3F.

6. Line 291- “These results suggest that all the DDR related pathways are involved in A3B regulation in myeloma cells.” This has not been testing in the presented study, there are dozens of DDR pathways, using a few ATR/ATM inhibitors does not prove the above statement.

We revised the sentence “These results suggest that all the DDR-PIKK pathways might be involved in A3B regulation in myeloma cells” as the reviewer suggested.

Various spelling mistakes through-out manuscript.

We checked the manuscript again and corrected minor mistakes and spelling errors.

Major points:

1. There is a lack of meaningful statistical analysis on many of the graphs making it difficult to determine whether the results are significant, single experiments or biologically relevant.

We added indications of statistical significance in Figs 3 and 4 and described in legends in the manuscript.

Reviewer #2: The authors demonstrate the creation of CRISPR/Cas9 knock in of 3x FLAG tag and IRES-EGFP to the end of APOBEC3B gene in myeloma cell lines that they subsequently use as a reporter for chemotherapeutic treatments that induce APOBEC3B expression. Using this system they were able to identify inhibition of ATM, ATR and DNA-PK suppressed EGFP expression in response to antimetabolite treatments in several myeloma cell lines. This work is important to the understanding of APOBEC3B in myeloma and the findings support what is already known in breast cancer regarding APOBEC3B and ATM/ATR inhibition.

The methods are well described and the reporter system well characterized in the figures.

Major issues: none

Minor issues: Methods section: Statistical analysis described but not reported anywhere in the text, figures, or figure legends. Suggest adding this information prior to publication.

 We added indications of statistical significance in Figs 3 and 4 and described in legends in the manuscript.

---

## [Decision Letter · Decision Letter 1]

17 Dec 2019

APOBEC3B reporter myeloma cell lines identify DNA damage response pathways leading to APOBEC3B expression

PONE-D-19-26273R1

Dear Dr. Takaori-Kondo,

We are pleased to inform you that your manuscript has been judged scientifically suitable for publication and will be formally accepted for publication once it complies with all outstanding technical requirements.

With kind regards,

Robert W Sobol, PhD

Academic Editor

PLOS ONE

Additional Editor Comments (optional):

Reviewers' comments:

Reviewer's Responses to Questions

**Comments to the Author**

1. If the authors have adequately addressed your comments raised in a previous round of review and you feel that this manuscript is now acceptable for publication, you may indicate that here to bypass the “Comments to the Author” section, enter your conflict of interest statement in the “Confidential to Editor” section, and submit your "Accept" recommendation.

Reviewer #1: All comments have been addressed

Reviewer #2: All comments have been addressed

2. Is the manuscript technically sound, and do the data support the conclusions?

Reviewer #1: Yes

Reviewer #2: (No Response)

3. Has the statistical analysis been performed appropriately and rigorously? 

Reviewer #1: Yes

Reviewer #2: (No Response)

4. Have the authors made all data underlying the findings in their manuscript fully available?

Reviewer #1: Yes

Reviewer #2: (No Response)

5. Is the manuscript presented in an intelligible fashion and written in standard English?

Reviewer #1: Yes

Reviewer #2: (No Response)

6. Review Comments to the Author

Reviewer #1: The authors have address all comments sufficiently. This reviewer thanks the authors for taking the time to improve the manuscript.

Reviewer #2: (No Response)

7. PLOS authors have the option to publish the peer review history of their article (what does this mean?). If published, this will include your full peer review and any attached files.

Reviewer #1: No

Reviewer #2: No

---

## [Editor Report · Acceptance letter]

23 Dec 2019

PONE-D-19-26273R1 

APOBEC3B reporter myeloma cell lines identify DNA damage response pathways leading to APOBEC3B expression 

Dear Dr. Takaori-Kondo:

I am pleased to inform you that your manuscript has been deemed suitable for publication in PLOS ONE. Congratulations! Your manuscript is now with our production department. 

With kind regards,

on behalf of

Dr. Robert W Sobol 

Academic Editor

PLOS ONE